# Prediction of Water Leakage in Pipeline Networks Using Graph Convolutional Network Method

Ersin Şahin [1,*] and Hüseyin Yüce [2]

1 Computer Programming, Beykoz Vocational School, Beykoz University, 34820 Istanbul, Turkey
2 Mechatronics Engineering, Faculty of Technology, Marmara University, 34722 Istanbul, Turkey; huseyin@marmara.edu.tr
* Correspondence: ersinsahin@beykoz.edu.tr

**Featured Application: Considering the theoretical contribution of the study to science, the use of graphs in monitoring leaks in pipelines and the application of graph-based machine learning for detection represent a novel approach in the literature. The datasets generated in this study will be made available to other scientists, serving as a foundation for further research and offering various benefits. When assessing the impact of this work on social life, it becomes crucial to utilize water resources effectively and efficiently due to increasing demand resulting from both global warming and urbanization. Ensuring the sustainability of our world heavily relies on this aspect.**

**Abstract:** This study aims to predict leaks in water-carrying pipelines by monitoring pressure drops. Timely detection of leaks is crucial for prompt intervention and repair efforts. In this research, we represent the network structure of pipelines using graph representations. Consequently, we propose a machine learning model called Graph Convolutional Neural Network (GCN) that leverages graph-type data structures for leak prediction. Conventional machine learning models often overlook the dependencies between nodes and edges in graph structures, which are critical in complex systems like pipelines. GCN offers an advantage in capturing the intricate relationships among connections in pipelines. To assess the predictive performance of our proposed GCN model, we compare it against the Support Vector Machine (SVM) model, a widely used traditional machine learning approach. In this study, we conducted experimental studies to collect the required pressure and flow data to train the GCN and SVM models. The obtained results were visualized and analyzed to evaluate their respective performances. The GCN model achieved a performance rate of 94%, while the SVM model achieved 87%. These results demonstrated the potential of the GCN model in accurately detecting water leaks in pipeline systems. The findings hold significant implications for water resource management and environmental protection. The knowledge acquired from this study can serve as a foundation for predicting leaks in pipelines that transport gas and oil.

**Keywords:** graph convolutional network; graph machine learning; leakage detection

## 1. Introduction

Due to the development of the industry, the need to transport water resources to cities through pipelines has arisen during urbanization, both from dams and underground sources. The demand for water resources is increasing due to population growth, while water resources are decreasing due to global warming. Given the increasing demand and reduced resources, it is essential to transport water resources without loss. Open or closed pipeline networks are used to transport and distribute water resources. Failures in pipeline networks are common and significant and a global problem. Failures in pipeline networks may occur in two ways: leaks and blockages. This study focuses on the detection of water leaks in pipeline networks. Pipeline leaks that arise from pressure or temperature changes,

corrosion, wear and tear, and third-party damages [1] may lead to critical problems in pipeline transportation systems. Accurately detecting water leaks in pipeline networks is essential for avoiding the economic impacts of losses and protecting the environment [2]. Therefore, it is crucial to detect and repair pipeline network failures rapidly. Timely and accurate intervention in pipeline failures is a critical decision-making problem. Two main approaches are used to detect failures in pipeline networks: physical inspection and mathematical model simulation. Physical methods are costly due to the production stoppage, although they accurately identify the location and size of the failure. On the other hand, the mathematical approach theoretically detects leaks and is, therefore, much less expensive [3]. Various methods are used to detect leaks in pipeline networks. Some of these methods include acoustic methods [4,5], negative pressure wave measurement [6,7], measurement of input and output values of pressure and fluid velocity [8–10], vibration analysis, distributed fiber optic sensing, infrared cameras, and lidar systems [11]. Many machine learning algorithms based on convolutional neural networks and artificial neural networks are used for leakage detection and classification [12]. Many studies have been conducted to detect and localize pipeline system leaks [13,14]. A growing body of literature on using machine learning applications for leak detection in pipelines [15,16]. However, studies on machine learning methods for leak detection in pipelines are still limited and have remained under-researched [17].

A literature review reveals that studies have been conducted on detecting leaks in pipelines using machine learning methods, which can be categorized into five main approaches. Machine learning models employed in the studies in the literature are as follows:

1. For the negative pressure wave (NPW) method, a wireless sensor network-based machine learning (WML) algorithm was utilized [18].
2. The acoustic-based model, including support vector machines (SVM) [19], neural networks [20], artificial neural networks (ANNs), as well as SVM [21], and the combination of SVM with Relevance Vector Machine (RVM) [22], was employed for leak detection methods.
3. The thermal infrared (IR) camera-based method utilized the immune neural network [23] and convolutional neural network (CNN) approaches [24] for leak detection. Some studies specifically focused on the application of these methods in petroleum pipelines.
4. Pressure-monitoring methods employed the sparrow search algorithm (SPSA) and CNN [25] for leak detection in petroleum pipelines, while a deep-learning method called Deeppipe was also used [26]
5. The fluid transient waves method utilized artificial neural networks (ANNs) [27] and a machine learning (ML)-based framework [28].

This academic literature review highlights the utilization of various machine learning algorithms to detect leaks in pipelines, which demonstrates the successful application of different methods and encourages further research to achieve improved results in leak detection techniques. Using machine learning algorithms, accuracy rates for pipeline fault diagnosis have been reported to range from 78.51% to 99%. Representing many complex systems encountered in real life theoretically can be challenging. These complex systems can be described using graph structures.

Many structures, such as airline, banking, social, medical, and supply chain structures, can be expressed using graph structures. It is feasible to depict intricate systems through graphs, such as distribution networks with multiple pipeline convergence or divergence points. In the literature, pipeline networks have been represented using graphs, and the spatial correlations of the pipeline network have been captured using the GCN model [29]. In this study, pipelines carrying water have been represented using graphs. The pressure values at certain points on the pipeline were visualized over time by modeling and observing pressure drops. This study used the graph-based machine learning algorithm GCN to detect pipeline leakage situations.

Pipeline networks are often not limited to a single pipeline; significantly, in systems like water distribution networks, there are structures where multiple pipelines converge or diverge. Representing these complex structures with graph data structures provides advantages compared to traditional data structures. Conventional machine learning models (SVM, ANNs) often overlook the inherent dependencies among nodes and edges in pipeline networks. However, it is crucial to acknowledge that a leakage occurrence in one pipeline can have cascading effects on interconnected pipelines. In light of this, this study proposes an approach that leverages graph data structures to effectively represent pipeline networks, accompanied by the utilization of graph convolutional networks (GCN) as a graph-based machine learning model. By incorporating GCN, the model can effectively capture and contain the interdependencies between nodes and edges, thereby enhancing the accuracy and predictive capabilities specifically tailored to the domain of pipeline networks. The studies on GCN in the literature generally focus on object recognition and classification. The GCN algorithm has achieved accuracy rates of 75% in urban pipeline networks [30], 94% in image processing [31], 88% in customer product recommendation [32], and 86% in fault detection on steam turbines [33].

This study aims to predict leaks by monitoring pressure drops in water-carrying pipelines, with a specific focus on accurately detecting pipeline leaks. In this research, we represent the network structure of pipelines using graph representations. The graph convolutional neural network (GCN) model is utilized. GCN offer an advantage in capturing the intricate relationships among connections in pipelines. The data required for the GCN algorithm were collected from a test set using experimental methods, wherein a leakage scenario was deliberately created. Two datasets, edge and node, were specifically constructed for this purpose. The performance of the GCN algorithm in leak detection was compared with existing studies in the literature. The primary objective was to evaluate the performance of GCN in graph-based machine learning and perform a comparative analysis with traditional machine learning methods, explicitly employing the Support Vector Machine (SVM) algorithm as a reference. Given the escalating demand for water resources and the concurrent decline in availability, preventing leaks in pipeline transportation systems has become paramount. Detecting leaks facilitates prompt intervention, maintenance, and repair, minimizing economic losses and mitigating environmental impact. The findings of this study highlight the efficacy of the GCN algorithm in water leakage detection and underscore the potential of graph-based machine learning in tackling intricate problems.

## 2. Materials and Methods

This study utilized the graph-based machine learning algorithms GCN and SVM. These algorithms were implemented using StellarGraph, Pandas, Numpy, Sklearn, Tensorflow, IPython, matplotlib, and Pyvis libraries based on Python. The performance evaluations of the GCN and SVM algorithms were conducted using a confusion matrix, accuracy, precision, recall, and f1 scoring methods. The GCN algorithms were implemented using early-stopping methods to prevent overfitting.

### 2.1. Graph Convolutional Networks

Graph Convolutional Networks (GCN) is a current artificial neural network research topic. The GCN model is derived from graph theory and convolution theorems to apply machine learning to data represented by graphs. In general, a node is represented by combining its attributes with the attributes of its neighbors [34]. GCN learns the representation of a node by propagating neighbor information based on the graph structure. A GCN is a multilayer neural network that operates directly on a graph and induces embedding vectors of nodes based on the properties of their neighborhoods. Formally, consider a graph $G = (V, E)$ [35], where $V$ and E are sets of nodes and edges, respectively [36].

The GCN algorithm was proposed by Kipf et al. [37], and its working principle for a single layer is expressed in Equation (1) below.

$$Z = \sigma\left(\widetilde{D}^{-\frac{1}{2}} \widetilde{A} \widetilde{D}^{-\frac{1}{2}} X W\right) = \sigma(\hat{A} X W) \tag{1}$$

$$\widetilde{A} = A + I \tag{2}$$

The expression $\widetilde{A}$ given in Equation (2) is the normalized adjacency matrix with self-loops. The degree of the normalized adjacency matrix is denoted by $\widetilde{D}_{ii}$ in Equation (3).

$$\widetilde{D}_{ii} = \sum_j \widetilde{A}_{ij} \tag{3}$$

"$\sigma$" in Equation (1) represents the activation function. In this study, tanh and Softmax functions were used as activation functions. "W" is the trainable weight matrix. "X" represents the input features of nodes and edges in the graph structure (pressure and flow rate values from sensors). "Z" is the output of the GCN layer containing first-degree neighborhood information of all nodes. If k layers are used in the GCN algorithm, the output of Z contains k-degree neighborhood (spatial) information. Therefore, the hidden-layer data of GCN can provide more preliminary information for model training, enabling trained hidden-layer neurons to have a more profound feature expression ability [38]. The difference between GCN and classical machine learning algorithms lies in the ability of GCN to directly operate on graph data structures and effectively utilize neighborhood information to model the graph structure. With these capabilities, GCN can better capture patterns and relationships in graph data, enabling more accurate predictions.

The application stages of the Graph Convolutional Networks (GCN) algorithm can be defined in 11 steps:

1. Initialize the initial node feature matrix $Z^0 = X$.
2. Determine the number of GCN layers (K) (16, 16).
3. Create weight matrices ($W^k$) for each GCN layer with random initial values.
4. Define an activation function ($\sigma$ = tanh).
5. Perform the graph convolution operation: $Z^{k+1} = \sigma(D^{-1} \times A \times Z^k \times W^k)$.
6. Apply additional steps for each GCN layer, such as dropout or normalization.
7. Create the output layer: $Z^{out} = softmax(Z^K \times W^{K+1})$.
8. Define the loss function (L), typically cross-entropy.
9. Choose an optimization algorithm (Adam optimizer).
10. Update the weights using the training data: $W^k$, $W^{K+1}$ = optimizer. minimize(L).
11. Make predictions on new data using the trained model: $Z^{out\_test} = softmax(Z^K \times W^{K+1})$.

$Z^k$ represents the matrix representing the node features; $W^k$ represents the weight matrix, A represents the adjacency matrix representing the graph structure and D represents the degree matrix. The algorithm updates the node features by utilizing the neighborhood information from the graph structure and uses the softmax activation function for classification. During the training stage, the weights are updated using an optimization algorithm, and during the testing stage, the trained model makes predictions on new data. These steps describe the implementation stages of the Graph Convolutional Networks (GCN) algorithm, including initializing the node features, determining the number of layers, creating weight matrices, applying the graph convolution operation, defining the output layer, defining the loss function, choosing an optimization algorithm, updating the weights during training, and making predictions on new data during testing. Table 1 below gives the parameters of the GCN model used in this study.

**Table 1.** Parameters of the GCN model.

| GCN Model Parameters | |
| --- | --- |
| Layer sizes | (16, 16) |
| Epochs | 70 |
| Optimization method | Adam (lr = 0.01) |
| Loss function | Categorical cross-entropy |
| Activation functions | Tanh, Softmax |
| Drop out | 0.4% |
| Model train/rest rate | 0.24% |
| Verbose | 2 |

### 2.2. Support Vector Machines

Support Vector Machines (SVM) are an efficient alternative to supervised classification [39]. The foundation of SVM was developed by Vapnik [40] and has gained acceptance due to its various attractive features and promising performance. The formulation embodies the structural risk minimization principle and is superior to conventional machine learning methods' traditional empirical risk minimization principle employed by conventional machine learning methods. The SVM was initially developed to solve classification problems [41]. During the training process, the SVM model was trained with the parameters test size 0.25 and random state 48. Table 2 below gives the parameters of the SVM model used in this study.

**Table 2.** Parameters of the SVM model.

| SVM Model Parameters | |
| --- | --- |
| C | 1.0 |
| Break ties | False |
| Cache size | 200 |
| Class weight | None |
| Coef0 | 0.0 |
| Decision function shape | ovr |
| Degree | 3 |
| Gamma | scale |
| Kernel | RBF |

### 2.3. Experimental Setup

This study conducted an experimental investigation to obtain the necessary datasets for the proposed GCN and SVM algorithms, aiming to predict pipeline leaks. The experimental design and implementation phase consisted of four main steps, which are provided below.

- Research Design
- Selection of Data Collection Methods
- Data Collection Process
- Data Recording

During the research design phase, the required data to be collected (such as pressure and flow rate) were determined. The experimental dataset was prepared to be ready for data collection following the created leak scenario, and the sensors were calibrated. The limitations and constraints of this research were identified.

The limitations of this experimental research are as follows:

- The system is assumed to be represented by eight pressure sensors and two flow meters.
- It is assumed that the experimental setup and measurement instruments are unaffected by external factors, such as heat, vibration, noise, and light.
- It is assumed that the characteristics of the region where the experimental setup is located, such as altitude, do not affect the precision of the measurements.

- It is assumed that the PVC pipe type used in the experimental setup represents other pipe types used in transmission lines.

    The limitations of the experimental research are as follows:

- The findings are limited to the data set obtained from the experimental setup.
- The representation of liquid fluids is limited to water, used as the fluid in the experimental setup.
- The pressure measurements are limited to the eight pressure sensors, two flow meters, and PVC pipes used in the experimental setup.

An experimental setup was designed to create data sets required to operate graph-based machine learning algorithms. A leakage scenario, which is detailed in the titled scenario, was made to represent possible leakage situations. Pressure and flow variables were measured over time through pressure sensors (MPS 500) and flowmeters using the experimental setup shown in Figure 1 for data collection.

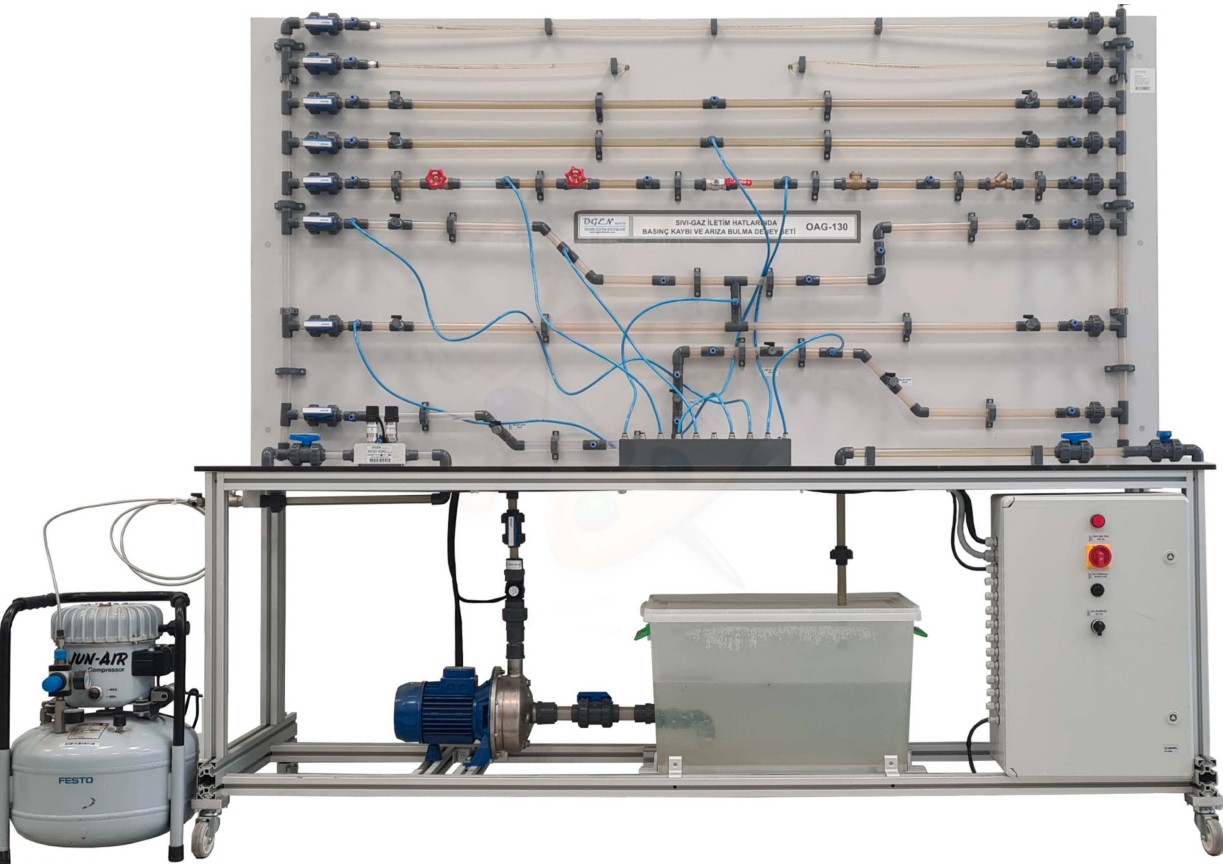

**Figure 1.** Experimental setup for pressure and flow measurement in the pipeline.

The data collection method and data recording processes were planned. During the selection of data collection methods, the nature and volume of the data were considered to determine how the data would be collected. In the data collection process, considerable attention was paid to collecting the data accurately and consistently. A 12-channel data acquisition card was used for recording the data. The data acquisition card records ten data points per second. The recorded data were transferred to electronic spreadsheets, and the accuracy and integrity of the data were checked. Any instances of incorrect or missing data were identified and corrected.

For graph-based machine learning algorithms to work, the obtained data must be represented by nodes and edges that connect these nodes. In the experimental dataset, the pressure sensors represent the nodes, and the connections between these sensors represent the edges. A total of eight pressure sensors were used in the dataset. Since the

measurement intervals of the variables used in the dataset (such as flow rate and pressure) differed, the dataset was standardized using the Python programming language and the Sklearn and StandardScaler libraries. Additionally, the dataset was normalized using the MinMaxScaler library, as artificial intelligence models perform more efficiently with values ranging between 0 and 1. The experimental setup includes faucets and valves at various locations for simulating leakage scenarios. Table 3 presents three columns: edge weight, source (node), and destination (node). The table contains a total of 855 rows representing the states of leakage (s3a), risk (s3o), and normal (s3n). The expression 's3n' indicates the absence of leakage or fault, 's3o' indicates the presence of a leakage risk, and 's3a' indicates the presence of leakage or fault. Edge weights are assigned as 3 for leakage scenarios, 2 for risky scenarios, and 1 for normal scenarios. The expressions A0, A3, . . . , A8 in Table 3 represent the pressure sensors in the experimental dataset, which are the graph nodes. The expressions _1, _2, _15 in Table 3 represent the values measured at different times corresponding to the respective nodes.

**Table 3.** Edge list.

| Source | Target | Edge Weight |
|---|---|---|
| s3n_A0_1 | s3n_A3_1 | 1 |
| s3n_A3_2 | s3n_A8_2 | 1 |
| s3o_A8_3 | s3o_A9_3 | 2 |
| s3o_A7_4 | s3o_A9_4 | 2 |
| s3a_A8_15 | s3a_A9_15 | 3 |
| s3a_A7_15 | s3a_A8_15 | 3 |

Table 4 details 360 rows representing the nodes. The node list has 339 columns representing the attributes of each node, such as pressure and flow rate. The node names are encoded as follows: s3n for the case of no leakage, s3o for the risky situation, s3a for the case of leakage, and A0_time for the zeroth node. For the leakage scenario, the node of the A0 sensor belonging to the normal state at the first second is represented as s3n_A0_1.

**Table 4.** Node list.

| Nodes | Feature-1 | . . . | Feature-339 | System State |
|---|---|---|---|---|
| s1n_A0_1 | 1 | . . . | 1 | Normal |
| s1n_A2_1 | 0 | . . . | 0 | Normal |
| s1o_A2_1 | 0 | . . . | 1 | Risky |
| s1a_A2_1 | 0 | . . . | 0 | Leakage |

## 3. Scenario

A scenario was created to represent leakage fault conditions in pipeline systems. Based on this scenario, the data collection process was carried out using the experimental dataset described above. Water was used as the fluid in the experimental dataset. Figure 2 shows the graph and the location of the leakage for the leakage scenario. The graph nodes (A0, A2, A3, A4, A6, A7, A8, A9) represent the pressure-measurement points in the experimental dataset. Pressure measurements were taken at eight different points. In the leakage scenario, the experimental dataset included four pipelines shown in green in Figure 2. Two different pipe diameters, 1 mm and 3 mm, were used in the transmission lines for the leakage scenario.

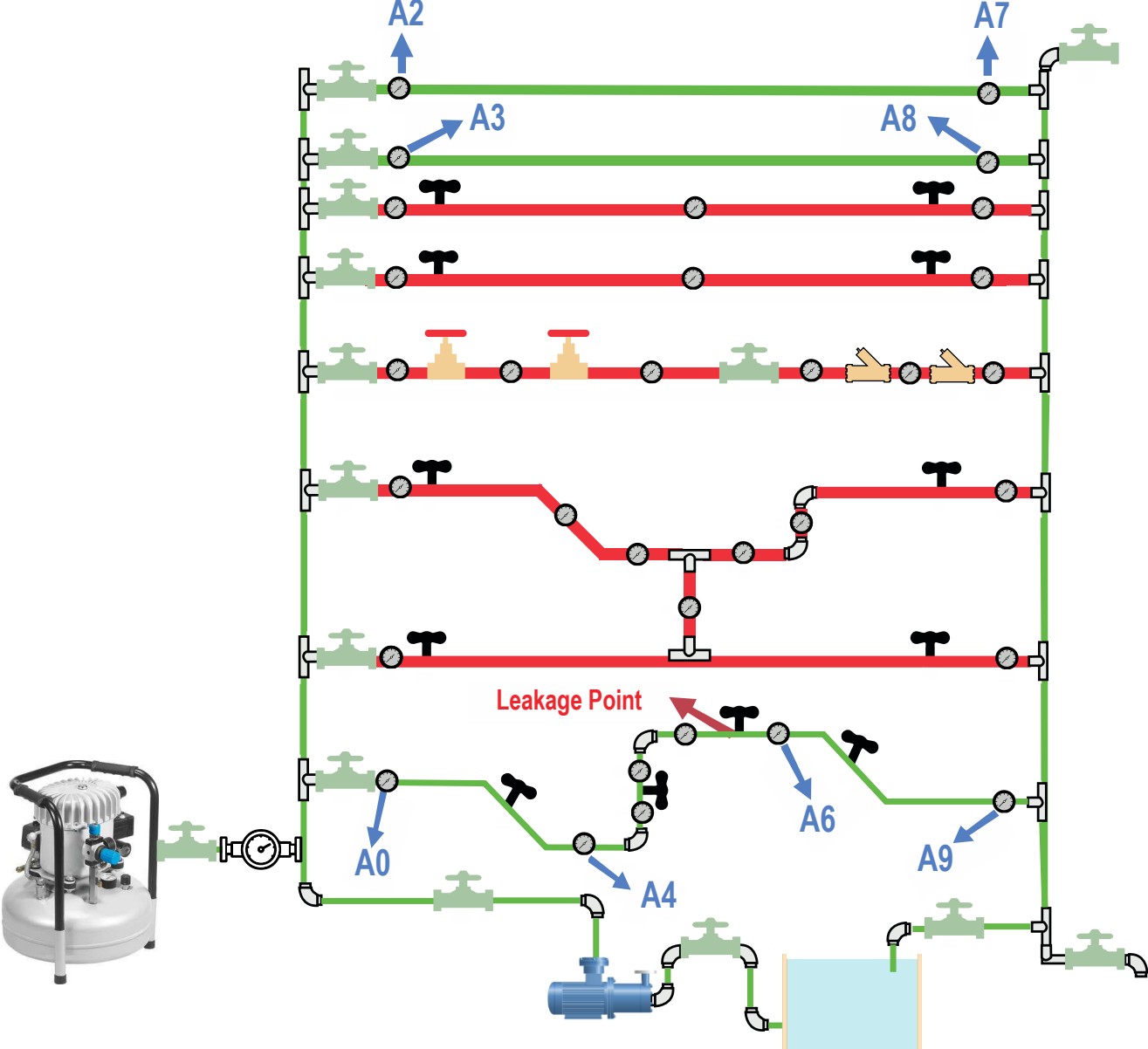

**Figure 2.** A scenario diagram was created for measuring the pipeline's pressure and flow rate.

The conditions of the created scenario are as follows:

1. The scenario was conducted at room temperature.
2. Before data collection, the air in the pipes was purged after operating the experimental setup.
3. The system was allowed to reach a stable state, and the pipes were completely filled with water before data collection.
4. The pressure sensors were calibrated.
5. Considering the structure of water distribution networks, pipes with different diameters and heights were selected for the scenario. The location of the leakage point was chosen so that measurements could be taken both before and after the leakage point using sensors.

The graph structure depicted in Figure 3 represents the pressure sensors and PVC water pipes in our experimental setup. The pressure sensors are the graph nodes, while the water pipes carry the edge attributes. The graph is directed, as there is a flow direction within the pipelines.

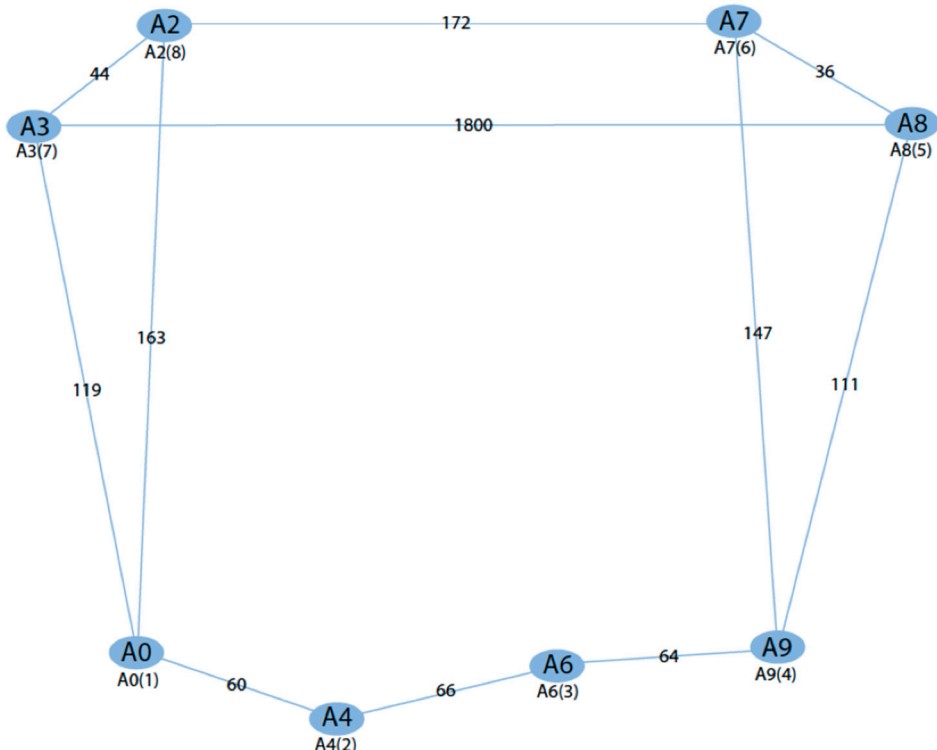

**Figure 3.** Graph representation of the scenario created for pressure and flow measurement in the pipeline.

The expressions A0, A2, A3, A4, A6, A7, A8, and A9 depicted in Figure 3 represent the pressure sensors present in the experimental dataset given in Figure 1. When the pipeline network in the experimental dataset is represented as a graph structure, the points A0, A2, A3, A4, A6, A7, A8, and A9 serve as the nodes of the graph structure. These node points are recorded by measuring the pressure values over time to create a dataset.

## 4. Results

Data collected with eight different pressure sensors and flow meters on the experimental setup were labeled for normal, risky, and leakage scenarios for a 10-s time interval and sampled in Table 5. The samples of risk and leakage are significantly different from most other samples regarding characteristics [42]. The data in Table 5 are visualized as a graph in Figures 4 and 5.

Each color seen in Figures 4 and 5 represents an individual pressure sensor within the experimental dataset, and their corresponding mappings are as follows: A0 (blue), A4 (red), A6 (orange), A9 (yellow), A3 (green), A2 (violet), A7 (brown), A8 (turquoise).

Using graphs, Figure 4 displays the experiment data (Table 5) representing normal, leak, and risky scenarios.

The central nodes represented in black show the time information. The colored nodes connected to the black nodes by edges, A0, A2, . . . and A9 represent the pressure information obtained from the sensors. The radii of the nodes vary according to the measured pressure values. In leak scenarios, i.e., in the first, seventh, and eighth central nodes, the node sizes are very small, as the pressure values are relatively lower than the normal scenarios. In risky scenarios, i.e., in the fourth and sixth central nodes, it is observed that the nodes shrink compared to normal scenarios. Figure 5 below illustrates the clustering of similar nodes (based on node and edge characteristics) using the Python NetworkX library for the values provided in Table 5, where the nodes are clustered closely together.

**Table 5.** (0–9) second experiment set data labeled as Risky, Normal, and Leakage (flow rate, pressure [mA]).

| Time | A0 | A2 | A3 | A4 | A6 | A7 | A8 | A9 | Flow 1 | Flow 2 | System State |
|------|------|------|------|------|------|------|------|------|--------|--------|--------------|
| 0 | 8.5852 | 7.7438 | 8.0905 | 7.9858 | 8.1753 | 9.9996 | 7.0367 | 9.9996 | 0.6433 | 0.6253 | Normal |
| 1 | 2.4679 | 2.7862 | 2.7090 | 2.7395 | 2.6995 | 4.5739 | 3.1051 | 4.9981 | 0.6365 | 0.6253 | Leakage |
| 2 | 8.6276 | 7.7267 | 8.0307 | 7.9724 | 8.1848 | 9.9996 | 7.0370 | 9.9996 | 0.6552 | 0.6298 | Normal |
| 3 | 8.6093 | 7.7267 | 8.0447 | 7.9504 | 8.1417 | 9.9996 | 6.9976 | 9.9996 | 0.6799 | 0.6179 | Normal |
| 4 | 5.0442 | 4.8980 | 4.9563 | 4.9505 | 4.9908 | 7.0755 | 4.7476 | 7.6266 | 0.6015 | 0.6021 | Risky |
| 5 | 8.5580 | 7.6980 | 7.9895 | 7.9409 | 8.1707 | 9.9996 | 7.0004 | 9.9996 | 0.6689 | 0.6253 | Normal |
| 6 | 5.0234 | 4.9099 | 4.9774 | 4.9642 | 5.0503 | 7.0767 | 4.7598 | 7.6596 | 0.6021 | 0.6024 | Risky |
| 7 | 2.5045 | 2.8204 | 2.7096 | 2.7218 | 2.7200 | 4.5855 | 3.1057 | 4.9801 | 0.6436 | 0.6253 | Leakage |
| 8 | 2.4459 | 2.8164 | 2.7236 | 2.7355 | 2.7081 | 4.5532 | 3.0987 | 4.9758 | 0.6668 | 0.6295 | Leakage |
| 9 | 8.5006 | 7.6895 | 8.0026 | 7.9452 | 8.1314 | 9.9996 | 6.9760 | 9.9996 | 0.6536 | 0.6195 | Normal |

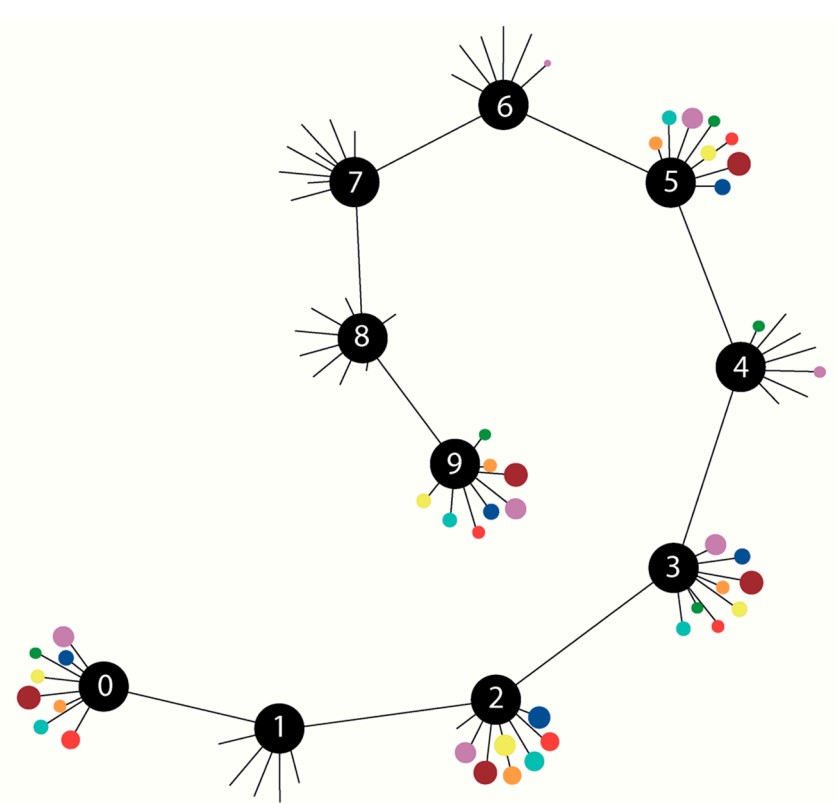

**Figure 4.** Graphical representation over time of the data provided in Table 5.

As shown in Figure 5, the sensor values obtained under normal conditions are clustered in a different area than those obtained during leaks and risky situations. The nodes in Figure 5 are numbered 1, 2, 3, etc. For each period in Table 5, eight sensor or node values are generated. For time 0, the node numbers are 1, 2, 3, ..., 8, and for time 1, the nodes continue from 9, 10, 11, ..., 16. The leakage situations in Table 5 are given at times 1, 7, and 8. It can be observed that the sizes of the nodes corresponding to these periods, which fall between nodes 9 and 16, 57 and 64, and 65 and 72, have shrunk to a size that is too small to be observed. The parameters related to the performance of the GCN algorithm are determined as an accuracy score of 0.9420, a loss value of 0.0980, and an f1 score of 0.94. The summary of the classification of scenarios according to the GCN algorithm is given in Table 6.

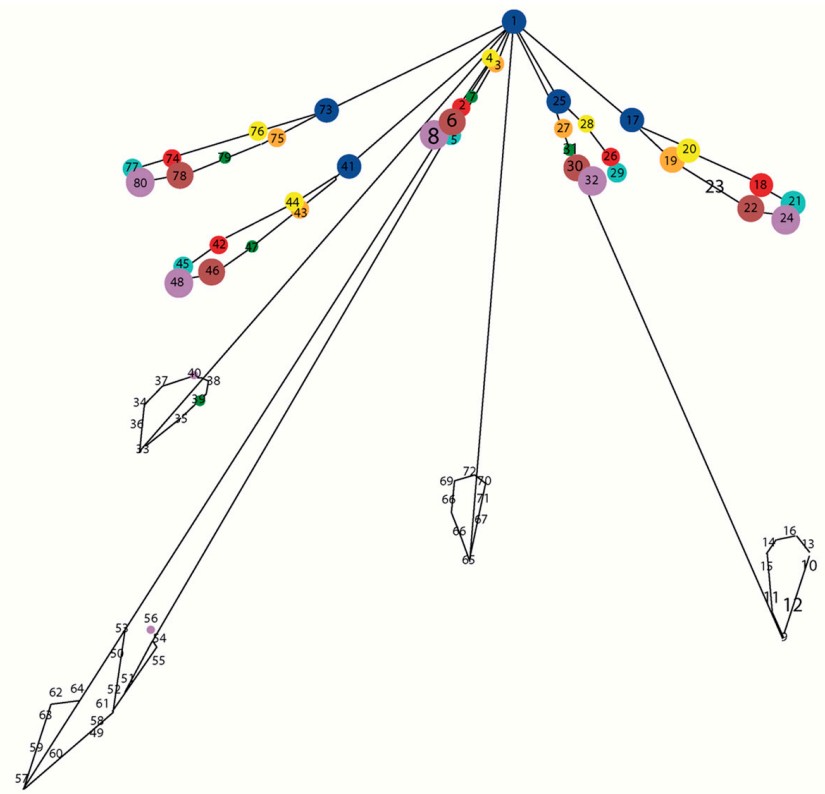

**Figure 5.** A clustered graphical representation of the pressure values of the sensors in Table 5, grouped according to their similarities.

**Table 6.** Summary of classification of scenarios according to the GCN algorithm.

| System Status | Precision | Recall | f1-Score | Sample |
|---|---|---|---|---|
| Normal | 1.00 | 0.82 | 1.00 | 120 |
| Risky | 1.00 | 1.00 | 0.94 | 120 |
| Leakage | 0.90 | 0.89 | 0.95 | 120 |
| Accuracy | | | 0.94 | 360 |
| Macro Average | 0.97 | 0.96 | 0.96 | 360 |
| Weighted Average | 0.97 | 0.96 | 0.96 | 360 |

For the leakage situations in Table 6, the f1-score is detected as 95%, the detection of risky situations with 94% accuracy, and the detection of normal situations with 100% accuracy. The system's average leakage, risk, and normal status are detected with a 94.20% accuracy rate. The real-time monitoring of leakage situations in pipeline networks and the detection of leakage situations occur at the moment of leakage. This situation causes a waste of time and water resources during the intervention phase of leakages. It is crucial to detect leakage situations in pipeline networks at an early stage. The system status being labeled as risky indicates the need for intervention in case of leaks. The prediction of a possible significant leakage situation is made in advance. Small interventions or maintenance and repair can be addressed at this stage before the failure escalates. The graph of the accuracy and loss values of the GCN algorithm is given in Figure 6.

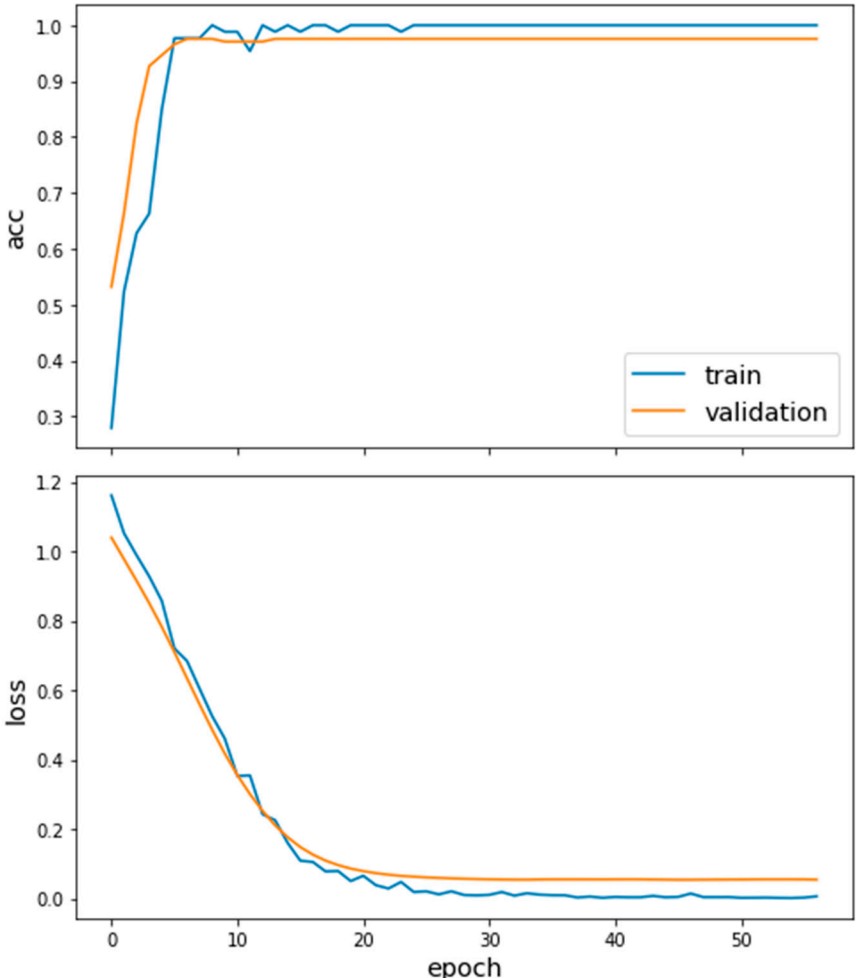

**Figure 6.** Accuracy and loss graph for the GCN algorithm.

The classification of the predicted values of the GCN algorithm for the system's risky, leakage and normal states is shown in Figure 7, which was created using the t-SNE (t-Distributed Stochastic Neighbor Embedding) technique, which is employed for visualizing high-dimensional data by assigning each data point a location on a two- or three-dimensional map [43].

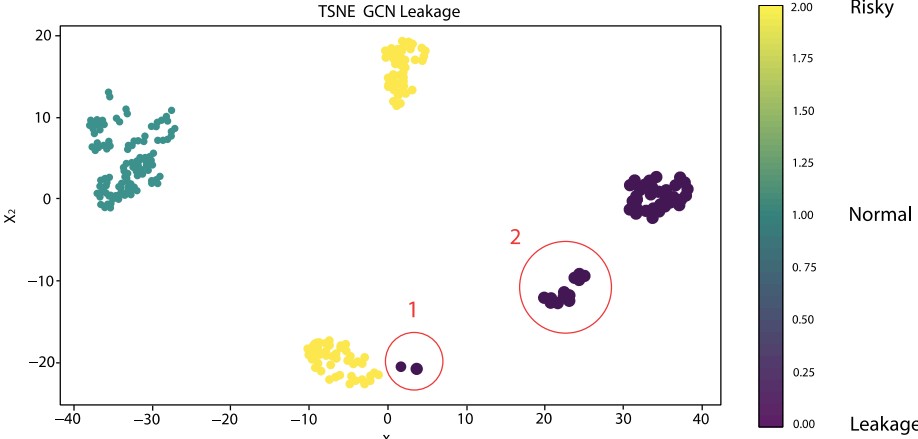

**Figure 7.** Classification of the system's risky, leakage, and normal states with GCN.

The $X_1$ and $X_2$ axes in Figure 7 represent the coordinates of the two-dimensional space generated by the t-SNE algorithm. The positions of the points on the graph are determined based on the similarities among the data points. The nodes labeled 1 and 2 within the circles in Figure 7 show the confusion of the algorithm in predicting leakage and risky situations, indicating that the nodes belonging to these situations have similar characteristics. The confusion matrix comparing the actual and predicted values of the GCN algorithm for the system's risky, leakage, and normal states is shown in Figure 8.

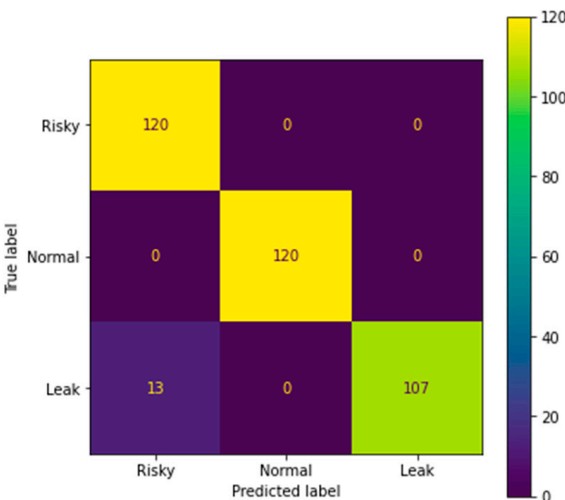

**Figure 8.** Confusion matrix according to GCN results.

The matrix compares the actual states with the predicted states obtained from applying the GCN algorithm in Figure 8. The findings showed that the system confused the risky situations with the leakage situations. The performance of the Graph Convolutional Network (GCN) algorithm, used for machine learning on graph-structured data, was evaluated by employing a classical machine learning algorithm, Support Vector Machine (SVM), for predicting leakages in pipeline systems using the same scenario and dataset. The parameters related to the performance of the SVM algorithm are determined as an accuracy score of 0.8666, a loss value of 0.1333, and an f1 score of 0.8667. The summary of the classification of scenarios according to the SVM algorithm is given in Table 7.

**Table 7.** Summary of classification of scenarios according to the SVM algorithm.

| System Status | Precision | Recall | f1-Score | Sample |
|---|---|---|---|---|
| Normal | 1.00 | 1.00 | 1.00 | 25 |
| Risky | 0.87 | 0.77 | 0.82 | 35 |
| Leakage | 0.76 | 0.87 | 0.81 | 30 |
| Accuracy | | | 0.87 | 90 |
| Macro Average | 0.88 | 0.87 | 0.87 | 90 |

For the leakage situations in Table 7, the f1-score is detected as 81%, the detection of risky situations with 82% accuracy, and the detection of normal situations with 100% accuracy. The system's average leakage, risk, and normal status are detected with an 87.66% accuracy rate. The SVM algorithm, which solely relies on node features and does not utilize edge features, has fewer samples than the GCN algorithm. The confusion matrix comparing the actual and predicted values of the SVM algorithm for the system's risky, leakage, and normal states is shown in Figure 9.

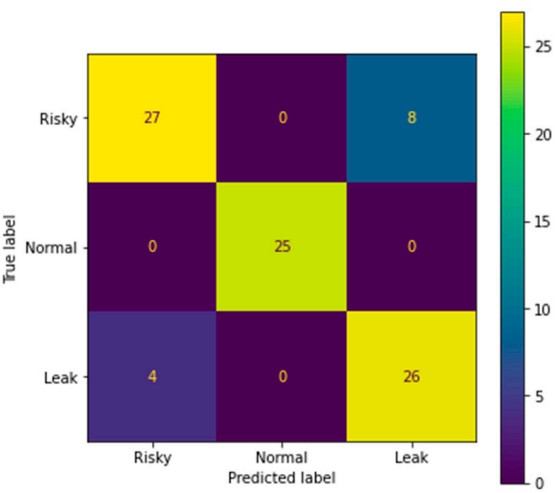

**Figure 9.** Confusion matrix according to SVM results.

The matrix compares the actual states with the predicted states obtained from applying the SVM algorithm in Figure 9. The findings showed that the system confused the risky situations with the leakage situations.

## 5. Conclusions

When the literature was reviewed, it was found that machine learning algorithms have been utilized for fault diagnosis in pipeline systems, yielding accuracy rates ranging from 78.51% to 99%. However, it is worth noting that non-graph-based machine learning models, such as SVM, ANNs, RVM, CNN, SPSA, and WML, have been predominantly employed instead of graph-based machine learning models for predicting and detecting leaks in pipelines.

In this study, the GCN model achieved a detection accuracy of 95% for leak conditions, 94% for risky situations, and 100% for normal situations. The SVM (Support Vector Machine) model was used as a reference to determine the GCN model's effectiveness. The SVM model achieved a detection accuracy of 81% for leaks, 82% for risky conditions, and 100% for normal conditions. Overall, when comparing the performance of the two models (considering leaks, risky conditions, and normal conditions), SVM achieved an accuracy of 87%, while GCN achieved 94% accuracy.

Based on the findings, the following conclusions have been drawn:

1.  The results reveal important implications for water resource management, reduction of water losses and sustainable living.
2.  The GCN model demonstrated high accuracy and performance in predicting pipeline water leaks, including leaks and risky conditions. The results highlight the effectiveness of the graph-based machine learning model. The high accuracy of the GCN model in detecting risky conditions is crucial for early leak detection and preventive maintenance planning. Situations labeled as risky indicate the need to act to prevent leaks. Thus, predicting a potential major leak can be made in advance. At this stage, it is aimed to solve the problem before it grows by performing small interventions or maintenance and repair.
3.  Both GCN and SVM models exhibited equally good performance in detecting normal conditions.
4.  The GCN model outperformed the SVM model when comparing the prediction results for leaks and risky conditions. The GCN algorithm's ability to analyze edge and spatial relationships made a difference in leak detection.

The results indicate the potential of graph-based machine learning methods in solving complex problems such as detecting water leaks in pipelines. Based on the experience gained from this study, it is aimed to replicate this study with other graph-based machine

learning algorithms, such as GraphSAGE, HinSAGE, RGCN GAT, SGC, PPNP, APPNP, and Cluster-GCN, given the detection of leak locations, and to predict leak and blockage situations in pipeline systems with gaseous fluids in further studies.

**Author Contributions:** Conceptualization, E.Ş.; formal analysis, H.Y.; investigation, E.Ş.; methodology, E.Ş.; project administration, H.Y.; writing—original draft, E.Ş.; writing—review and editing, H.Y. and E.Ş. All authors have read and agreed to the published version of the manuscript.

**Funding:** This study was supported by a scientific research project numbered "FDK-2023-10459" by Marmara University within the scope of the doctoral thesis.

**Institutional Review Board Statement:** Not applicable.

**Informed Consent Statement:** Not applicable.

**Data Availability Statement:** The data that support the findings of this study are available upon reasonable request from the corresponding author.

**Acknowledgments:** The experimental stages of this study were conducted using the experimental setup developed for this study at the Fault Diagnosis Laboratory of Marmara University, Faculty of Technology, Mechatronics Engineering Department in Turkey. I am indebted to Marmara University for providing the facilities to conduct the present study.

**Conflicts of Interest:** The authors declare no conflict of interest.

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
