# Peer review of "Prediction of Water Leakage in Pipeline Networks Using Graph Convolutional Network Method"

_applsci, doi:10.3390/app13137427_

Round 1
Reviewer 1 Report
This paper proposes a GCN based model to predict water leakage in pipeline. The model can be applied in practical processes. In general, the structure of this paper is clear, and the theory is presented in an easily understandable way. The following are some comments that can help to improve the manuscript.
1. The clarity of some figures can be improved.
2. How to determine the graph structure according to the prior knowledge and data? The authors should illustrate the methodology.
3. Please discuss other conventional deep models and carry out experiment to prove the effectiveness of the GCN based model.
4. How is the model trained? It is also not clear how the testing and validation are done. More detailed descriptions are needed including the model hyperparameters.
Minor editing of English language required.
Reviewer 2 Report
This paper shows Prediction of Water Leakage in Pipeline Networks Using Graph Convolutional Network Method. I have following suggestions: 1) The background to Prediction of Water Leakage should be clearly shown. What are the new features to your method? You had better to give Advantages and disadvantages of existing methods, why or form which view you start your research point. 2) In the abstract part, you had better to show your method in step by step in order to show the core work you have done. Form the current abstract, I cannot find the innovation, which is superficial to tell your method. You had better to add the more step details. In introduction part, you have to reconstruct it according to the sequence of background, problem to state,existing methods with disadvantage,your method and from which point to solve. I find the related works are disorder. You can cite DOI:10.1109/TNNLS.2022.3155486. DOI:10.1109/TNSE.2022.3163144. DOI:10.1108/IJWIS-04-2022-0081. DOI:10.1108/IJWIS-04-2022-0078 It should be sorted based on different method or methods. After you give and discuss proposed methods,you should state the problem or summary. 3) The experiment should be improved a lot. Please recheck experiment section that the experiment is hard to read. you have to reconstruct it according to the sequence of experiment target,Data preparation, how to compare?and your metrics. You should give the step of experiment that how to carry out, such as parameter,step, condition in scenario. You should show the configuration, that how to compute? You should reconstruct it according to the sequence of Phenomena, causes, and recommendations. You should give more details and Analysis according to experiment above you carry out.
It can be improved.
Reviewer 3 Report
This study aims to address the crucial issue of accurately detecting and predicting pipeline leaks in the context of increasing water resource demand and decreasing availability due to global warming. The authors explore the application of the graph convolutional neural network (GCN) algorithm to predict water leakage in pipelines. It provides a compelling rationale for the study by linking the increasing demand for water resources and the critical problems caused by leakage in pipeline transportation systems.
The creation of experimental data sets and the evaluation of the GCN algorithm's performance demonstrate a rigorous approach to the research. The manuscript highlights the average performance of the GCN algorithm (94.20%), establishing its effectiveness in predicting water leakage in pipelines carrying water. Here are some comments below:
The manuscript could benefit from providing more detailed explanations of the graph convolutional neural network (GCN) algorithm and its implementation. This would enhance the clarity and understanding of the methodology for readers who may not be familiar with this specific approach.
To further strengthen the study, it would be valuable to include a comparative analysis of the GCN algorithm with other existing methods or approaches for predicting pipeline leaks. This would provide a broader perspective on the algorithm's performance and its advantages over alternative techniques.
Overall, this paper successfully addresses the importance of accurately detecting pipeline leaks in the context of increasing water resource demand. By utilizing the graph convolutional neural network (GCN) algorithm, the study demonstrates a promising approach to predict water leakage in pipelines carrying water. This study contributes to the field of pipeline failure detection and provides a foundation for future research endeavors.
The quality of English language used in this paper is generally good. The manuscript is well-written and effectively communicates its ideas and findings. The sentences are clear and coherent, and the overall structure of the paper is well-organized.
Round 2
Reviewer 2 Report
I vote to accept this paper
It can be improved.